# Transposable elements contribute substantially to naturally occurring genetic lethality in *Drosophila melanogaster*

Sarah B. Marion[1]*, Katrina Focht[1], Iman Hamid[1], Edwin S. Iversen[2], Hannah John[1], Brenda Manzano-Winkler[1], Amber Navarra[1], Saniya Pangare[1], Mehrnaz Zarei[1], Mohamed A. F. Noor[1]

**1** Biology Department, Duke University, Durham, North Carolina, United States of America, **2** Department of Statistical Science, Duke University, Durham, North Carolina, United States of America

* msarah@reed.edu

## Abstract

Recessive lethal mutations are widespread across studied species, with estimates suggesting that each individual carries at least one. Numerous lethal alleles persist in wild populations at higher frequencies than expected given their extreme deleterious nature. Though these findings spurred historical debate whether classical balancing selection maintains some lethal alleles at elevated frequencies (versus mutation-selection balance acting alone), we propose the question remained unanswered, especially given that the genetic basis of most naturally occurring lethal effects is still unknown. Given current genome-wide point mutation rate estimates, mutation-selection balance alone cannot explain some of this lethal variation in nature. However, evolutionary biologists have historically studied genetic variation through a lens of single-nucleotide variants, when in fact the spectrum of mutational changes is far broader than point mutations alone, including indels, structural variants, short tandem repeats, and transposable element insertions. We uncover the genetic basis of lethality in nature and provide insight on the possible evolutionary forces allowing some to persist at higher frequencies. By locating hundreds of recessive lethal mutations in *Drosophila melanogaster* via complementation testing, fine-mapping, and sequencing a subset, we determine candidate lethal mutations in specific genes. We discover that many lethal disruptions are likely caused by transposable element insertions. The most common transposable elements in our data, *Transib1* and *Kuruka*, are both estimated to have recently invaded *D. melanogaster*, each from a different Drosophila species (between 2013–2016 and 2017–2021, respectively). This finding demonstrates that the many lethal alleles studied in *D. melanogaster* in the last century had a distinct genetic basis. Hence, we propose a model that could explain lethal variation in natural populations of *D. melanogaster*: lethal mutation frequencies are driven by invasions of new transposable elements and as time passes after each invasion, those frequencies decline as *D. melanogaster* evolves suppression mechanisms,

**Data availability statement:** Whole-genome long-read sequencing data from balanced lethal lines are publicly available in the NCBI Sequence Read Archive (SRA) under BioProject accession PRJNA1416996, with SRA run accessions SRR37072411-SRR37072452. All other relevant data (including fasta sequences of transposable elements) are available in the Supporting information files.

**Funding:** This work was supported by NSF grant DEB 2019789 to MAFN. The funders had no role in study design, data collection and analysis, decision to publish, or preparation of the manuscript.

**Competing interests:** The authors have declared that no competing interests exist.

**Abbreviations:** LOF, loss-of-function; LTR, long terminal repeat; MR, male recombination; TEs, transposable elements; WGS, Whole-Genome Sequencing.

allowing for natural selection to more efficiently remove lethal insertions. Upon the invasion of a new TE, the cycle repeats. The ubiquity of lethal alleles in natural populations is a classic conundrum for evolutionary geneticists for over a century, and this study utilized modern tools and sequencing technology to provide novel insight into this age-old mystery.

## Introduction

For nearly a century, evolutionary biologists have detected evidence of naturally-occurring, recessive lethal mutations in a wide taxonomic breadth of species, including mammals (*Homo sapiens*, M*us musculus*, *Sus scrofa domesticus*), insects (*Tribolium castaneum*, *Drosophila*), fish (*Lucania goodei, Danio rerio*), amphibians (*Xenopus laevis*, *Triturus cristatus carnifex*), and plants ((*Dactylis glomerata, Isotoma petraea*) [1–7]). Moreover, humans, African clawed frogs, zebrafish, bluefin killifish, and fruit flies have all been estimated to carry between 0.58 and 1.9 lethal mutations per individual [1,8–12]. The consistency of these figures across taxa is particularly remarkable given that researchers have used widely varying estimation approaches across studies. Many lethal allele frequency estimates have focused on humans, but these studies tend to use known disease-causing mutations or population genetic approaches that infer lethality based on underrepresented genotypes. The latter approaches are broad and agnostic, but they are inferential and can be affected by local demographic or other parameters. The most robust datasets on naturally occurring lethal effects are collected from *Drosophila melanogaster* due to their ease of experimental manipulation, short generation times, and the availability of the unique genetic tool known as a balancer chromosome. Drosophila balancer chromosomes allow testing of complete, intact wild chromosomes for homozygous lethal effects, and research has consistently shown that, similar to other organisms, lethal mutations are common in wild *D. melanogaster*, with at least 25% of chromosomes found to be homozygous lethal before adulthood (reviewed in [13]).

Beyond the ubiquity of lethal mutations in general, certain lethal alleles are detected at unexpectedly high (>1%) frequencies in natural populations. A chlorophyll-deficient lethal was found to exist between 2% and 15% in wild populations of orchard grass, and a specific lethal allele persisted at 1% frequency over an 8-year span in a *Drosophila melanogaster* population [14,15]. Some unusual cases are even more extreme. Balanced lethal systems exist, such as in crested newts, where 50% of all offspring die during development because there are only two homologous haplotypes present, each fixed for a private lethal allele, such that only heterozygotes survive [16]. The low effective size and selective pressures associated with domestication can lead to the persistence of lethal mutations at high frequencies in livestock populations; for example, a recessive lethal deletion has been maintained at 5% frequency due to heterozygote advantage in a commercial pig population [2,17].

Higher frequencies of some lethal mutations in Drosophila sparked historical debate surrounding the evolutionary forces that could result in such frequencies. All

deleterious mutation frequencies are influenced by negative selection removing them, new mutations adding them back again (mutation-selection balance) and the random chance of genetic drift. The expected equilibrium frequency of a recessive allele (in an infinite population) is dependent on the mutation rate ($\mu$) and the selection coefficient (s) [18,19].

$$q_l = \sqrt{\frac{\mu}{s}}$$

Given average single-nucleotide Drosophila mutation rates are on the order of $10^{-9}$ per base pair per generation [20–23] and a lethal selection coefficient is $s = 1$, mutation-selection balance predicts individual lethal mutations caused by single base changes should be very rare and found likely only once per sample (on the order of $10^{-5}$). Mukai and Crow estimated the Drosophila lethal mutation rate to be 0.0060 per whole chromosome per generation, with other mutations with smaller effect arising at 10–20 times as often. This figure could even be an over-estimate considering the observed mutation rate was likely inflated due to hybrid dysgenesis in the laboratory due to p-elements present in wild flies and not in the lab stocks to which they were crossed [24,25]. While many lethal mutations are as rare in natural populations as may be expected by mutation-selection balance based on these mutation rates, cases of high-frequency lethal effects cannot be easily explained by this model. During the 1930s through 1970s, when wild lethal mutation frequency was widely studied, some researchers hypothesized that balancing selection could play a role in driving lethal mutations to higher-than-expected frequencies. However, fitness assays in the laboratory generally failed to detect fitness benefits to being a carrier of wild-derived lethal chromosomes, and balancing selection as an explanation was broadly dismissed [26].

Uncovering the causal mutations of lethal effects in nature, an interesting question in its own right, is key to generating informed hypotheses regarding the evolutionary forces maintaining lethality in nature. Given the universality of genetic lethal effects across the tree of life and that research on recessive lethal mutations dates almost a century, we know surprisingly little about the genetic basis of lethality in nature, let alone the evolutionary forces that maintain those that are sampled at measurable frequencies. Historically, researchers have assumed that recessive lethal effects are largely due to single locus, loss-of-function mutations rather than polygenic effects, yet this hypothesis has never been tested directly. If recessive lethal phenotypes are due to single locus effects, then what are the genetic bases of these mutations with extreme effects? Beyond human-disease-causing mutations, only a handful of lethal mutations have been definitively identified (e.g., the 212kb deletion in the BBS9 gene in commercial pigs), and some have been inferred from sequence data [8], but no study has yet systematically and directly assayed the genome for naturally occurring lethal mutations. Outside of human disease research, even less is known about the classes of genes in which lethal mutations occur or their distribution throughout the genome.

Many researchers focus on single-nucleotide substitutions as the major driver of evolutionary change [27]. However, genomes are often shaped by other mutation types, such as structural variants, copy-number variations, and the movement of transposable elements (TEs). Because mutation rates of short tandem repeats mutations and TE transposition rates are typically higher than single nucleotide changes, they may be particularly effective agents of evolutionary change [27]. Many TE insertions have deleterious impacts on the host genome, and in fact, more recent research has uncovered unique adaptive impacts of TE insertions may confer adaptations in multiple species [28–30].

With modern genomic mapping tools and sequencing technology, we can now uncover the genetic basis of naturally occurring lethal effects and address multiple outstanding questions in evolutionary genetics. First, testing if naturally occurring lethal effects are due to single mutations touches a fundamental dichotomy in evolutionary genetics: which evolutionary processes are acting on phenotypic variation resulting from few mutations with large effects versus those resulting from many mutations with small effects? Second, if single mutations are the cause of most lethal effects, determining the underlying mutations and genes they disrupt will provide both novel insight into distribution of mutation types that cause the most extreme deleterious effects in the genome and inform our hypotheses regarding evolutionary forces responsible for high overall and individual lethal allele frequencies in nature.

To address outstanding questions regarding the genetic basis of natural lethal variation and provide foundational data to explore forces maintaining some lethal effects at higher frequencies, we survey the genetic basis and population frequencies of naturally occurring lethal mutations along most of the second chromosome in a large sample of wild *Drosophila melanogaster* collected in Durham, North Carolina, USA between 2018 and 2021. The second chromosome has the most widely used balancer chromosome available and is the largest of four chromosomes in the *D. melanogaster* genome. Previous studies on wild *D. melanogaster* have been conducted on populations from central NC [31], so Durham is an excellent population in which to examine these questions. Using the second chromosome balancer *CyO*, we generate 293 distinct lines bearing lethal mutations (hereafter, "lethal lines"), each heterozygous for a distinct wild second chromosome that is lethal when made homozygous. We perform an extensive scan of each lethal chromosome to localize the genetic basis of lethal effects, test if most naturally occurring lethal effects are due to single locus and loss-of-function mutations, and examine the distribution of lethal effects along the chromosome. We then fine-map a subset of lethal alleles to a single gene and use whole-genome sequencing to determine the precise regions disrupted and characterize lethal lesions at the sequence level. Our methods and results are presented using the following framework of guiding questions:

1. What causes so many *D. melanogaster* chromosomes to result in death before adulthood if homozygous? Is the underlying genetic basis of such lethality a result of single loci of large effect or combinations of small fitness effects?

2. Are there certain genes disrupted by lethal mutations at higher frequencies than expected by mutation-selection balance?

3. Are lethal mutations that disrupt the same gene identical by descent, or do different mutations break the same gene, potentially suggesting unusually "fragile" genes or gene regions?

4. Is there an overall trend in lethal mutation types and types of genes? Do we see any possible differences in mutation-types or genes between singletons and genes that are more frequently disrupted by lethal mutations?

5. Does the distribution of lethal mutations correspond to any features of the chromosome landscape such as gene density, recombination rate, or repetitive element density?

## Results

### Lethal chromosomes resulting from multi-locus or single-locus effects

To identify genes bearing lethal mutations and patterns among them, we generated a pool of what we term "lethal lines" that each carries wild-derived chromosomes that are lethal when made homozygous. This approach has additional benefits in that decades of work have been conducted on the frequency of such lethal chromosomes (including even local to the populations we sample) and can be used as comparison points for overall lethal chromosome frequencies. Out of the 555 second chromosomes sampled (sublines), we find that 47% (262/555) are lethal when made homozygous (Fig 1). See methods for lethal chromosome sample size and frequency calculations.

The ability to map lethal alleles to a distinct region is a proof-of-principle that at least some lethal mutations are single-locus and result in LOF, since both conditions must be true for a lethal allele to be located via deficiency mapping. We completed 19,127 crosses between 293 balanced lethal lines each to the majority of 70 balanced deficiency lines with an average coverage of 67% of the second chromosome. 250 distinct lethal mutations were mapped across 169 of the 293 lethal lines. Many lethal lines failed to complement with multiple distinct deficiencies, suggesting that many wild chromosomes bear multiple lethal mutations (Fig 2). At least one lethal mutation mapped to 59 out of 70 the deficiency regions.

Previous arguments have been made that deleterious mutations identified through deficiency mapping may represent haplo-insufficient mutations [32]. A haplo-insufficient lethal mutation would reduce the amount of protein product such that

**Wild 2nd Chromosomes Lethal vs Non-lethal collected in Durham, NC (2018-2021)**

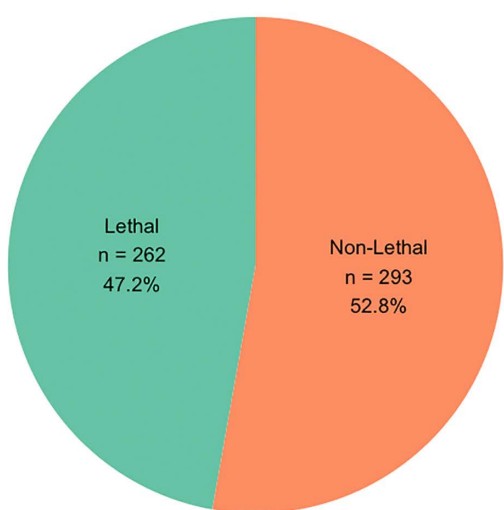

**Fig 1. Proportion of wild second chromosomes that are lethal when made homozygous from Durham, NC (2018 −2021).** 555 distinct second chromosomes were sampled from a wild population of D. melanogaster in Durham, NC between 2018 and 2021. Underlying data can be found in Table A in S1 Data.

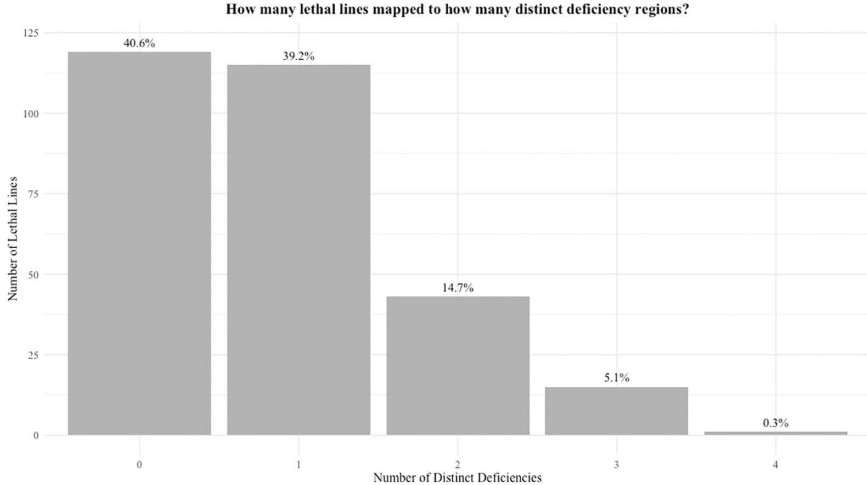

**Fig 2. Proportions of lethal lines that have failed to complement with 0, 1, or more distinct regions of the second chromosome.** This figure summarizes mapping data from all lethal lines ($n = 293$). If two deficiencies overlapped with one another and the same lethal line mapped to both, we only counted this as mapping to one distinct deficiency to be conservative in our estimates. Any lethal lines that mapped to more than one distinct deficiency are interpreted to mean that the wild chromosome they carry has more than one distinct recessive lethal mutation. Those that did not map to any deficiencies either carry lethal mutations in the regions not covered by the suite of deficiencies used (only cover at maximum 70% of the second chromosome) or the lethal mutations they carry are not able to be mapped through deficiency crosses (e.g., gain-of-function or multigenic synthetic lethals). Underlying data can be found in Table B in S1 Data.

heterozygotes for the mutation and a null mutant do not survive, but homozygotes produce enough protein to survive. The persistence of the deficiency stocks we used supports the idea that the loci surveyed are not inherently haplo-insufficient,

but does not disprove that some of the specific wild alleles at those loci may have been haplo-insufficient. If some of the lethal mutations we mapped were haplo-insufficient, then in principle, such alleles could be detected in chromosomes that are not lethal when made homozygous. To determine if we could identify any haplo-insufficient lethal mutations, we crossed 10 non-lethal lines (*CyO/+*, + being a non-lethal chromosome isogenized from wild flies collected from same populations as the lethal lines) to the 63 of our deficiency stocks and no lethal effects were mapped (the other 7 deficiency lines were obtained after completion of haploinsufficiency testing). While this experiment did not identify any haploinsufficient lethal mutations, it does not indicate that none exist. However, taken together, our mapping data is most naturally explained by homozygous lethal mutations rather than haplo-insufficient alleles.

The ability to map some lethal mutations suggests that at least some of the mutations we localized are LOF and possibly single locus. But is there evidence that *most* naturally occurring lethal mutations are single-locus and LOF? A null hypothesis was generated to estimate the number of lethal mutations that would be expected to map if they were all single locus and LOF (i.e., the two requirements that allow a lethal mutation to be mapped via deficiency crosses). It assumes a random distribution of lethal mutations along the chromosome and that some chromosomes would carry more than one lethal mutation (with the number of lethal mutations per chromosome following a Poisson distribution). Considering the number of lethal chromosomes tested and that on average 67% of each chromosome is scanned for lethal mutations, the null model estimates that 238 lethal mutations would be found (95% CI: 230–246). The estimate of the Poisson mean utilizes the frequency of lethal chromosomes in our sample (262/555). Of the 262 lethal lines, 232 distinct lethal mutations mapped, falling within the 95% CI of the null hypothesis.

### Sample frequencies of allelic lethal sublines (how frequent are lethal lesions in the same gene?)

To identify lethal gene disruptions that are present in more than one wild fly, we focus on sets of lethal lines that map to the same deficiency. Of the 59 deficiencies to which at least one lethal mapped, 50 deficiencies have more than 1 lethal line map to them. Of these lethal lines, we performed over 1,000 allelism crosses between lethal lines that mapped to the same region to determine which lethal lines contained mutations in the same gene (these lethal lines are from here on out called "allelic sets"). We found 35 allelic sets, ranging in frequency from 0.3% to 1.3% of the wild-sampled second chromosomes (Fig 3). Of the 5 allelic sets from 5 or more wild flies (~1%), 4 of these sets contain wild flies collected in different years, suggesting that lethal mutations in multiple genes have persisted or re-emerged across the 3.5-year span of sample collections (Fig 3).

### Genetic basis of abundant lethal genes: Are lethal mutations that disrupt the same gene identical, or do different mutations break the same gene?

We fine-mapped and sequence a total of 41 lethal sublines, including those in the 5 largest allelic sets (allelic sets >5 sublines) and some lower-frequency and singleton lethal mutations. In the largest allelic sets, we localized the lethal effect to a single gene in 4 out of 5 allelic sets (Table 1). WGS revealed candidate lethal mutations in all 4 genes identified through fine mapping (Table 1). All candidate mutations reported (with the exception of two detailed below) were not present in any sequenced non-allelic lethal lines. The first set of 6 allelic sublines mapped to *nipped-A*, a gene involved in DNA repair and Notch-signaling in wing development [33]. Each of the six sublines (from flies collected across all three years) failed to complement with *nipped-A* knock-out (*BDSC #16514*), and candidate lethal mutations were found within *nipped-A* sequence data. Five lethal lines carry TE insertions in the same 1,500 base-pair window in exon 16. All 5 insertions match closely with the "cut and paste" DNA transposon, *Transib1* and have the characteristic 5 base pair target site duplication (TSD). The sixth lethal mutation is 414 bp deletion in exon 19.

All 9 sublines in the second allelic set (from both 2018 and 2021 samples) mapped to the gene *lethal (2) giant larvae* (*l(2)gl*) (knockout obtained from the Lai Lab at Sloan Kettering Institute). *L(2)gl* is a tumor suppressor gene and closely related homologue to human tumor suppressor genes *Hugl-1* and *Hugl-2* [34]. *L(2)gl* is located in the

 

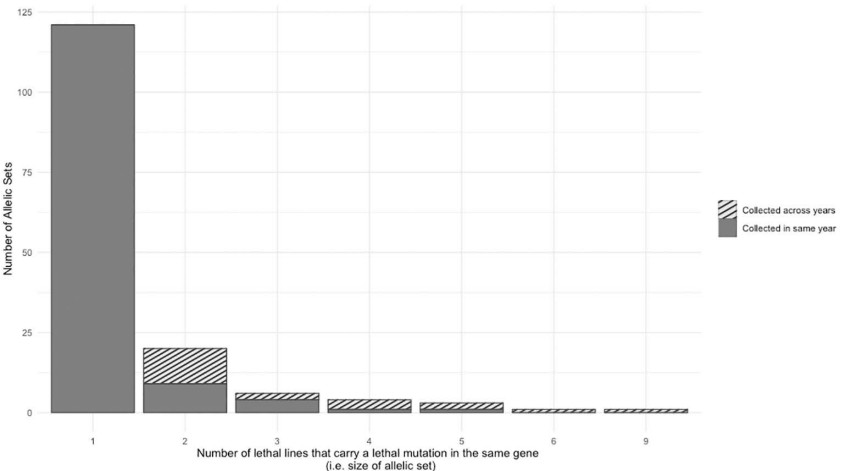

**Fig 3. Distribution of allelic set sizes and collection years.** Genes with lethal mutations that appeared only once in our sample are shown as allelic set size of 1 (i.e., a singleton). Allelic set sizes of 2 or more means that 2 or more lethal lines carry lethal mutations in the same gene. The proportion of allelic sets (of 2 or more lines) that consist of at least two wild flies sampled from different years is shaded. Underlying data can be found in Table C in S1 Data.

peri-telomeric region of 2L, less than 10kb away from the end of the reference genome. Sequence data suggests that lethality in these sublines is a result of a large telomeric deletion that fully or partially deletes *l(2)gl*. Breakpoints estimated from phased bam files differ between sublines, ranging from ~13kbto ~24kb away from the end of the reference genome (see S1 Fig).

Five sublines, all collected in 2018, contained lethal mutations in *drosha,* a gene involved in oocyte development and miRNA processing (from the Lai Lab at Sloan Kettering Institute). All sublines carried the identical premature stop codon (C to G) candidate lethal mutation in a *drosha* exon. As all 5 sublines that mapped to *drosha* were collected within 2 weeks of one another, we performed a kinship analysis to determine the relative relatedness of the sequenced lethal lines. As all lethal lines carry the same balancer chromosome, absolute kinship coefficients are elevated. Relative to the relatedness between other lethal lines, 4 of the 5 lethal lines that map to *drosha* appear to be related, which together with the closeness in date when the original flies were collected could explain this case of a high frequency of an IBD lethal allele in our sample (Fig 4).

The fourth allelic set includes 5 sublines (from flies collected in all three years) that all mapped to *Ca-alpha1D* (*BDSC #13282*)*,* two of which were sequenced, and each carried a unique high-impact mutation in *Ca-alpha1D*. One mutation is a pre-mature stop codon and the other a SNP at a splice site donor. Both lines also happened to carry a unique inversion that inverted the entire gene, with breakpoints ~20kb upstream and ~15kb downstream of *Ca-alpha1D*.

The fifth set (composed of 5 allelic sublines from all three collection years) was more challenging to narrow down. One of the 5 sublines was also allelic to two different sublines (from 2018 to 2021) and all three mapped to the gene *short stop* (abbreviated *shot: BDSC #10522)*, an important gene in cytoskeletal organization (Table 2). Two sublines from different years contained TE insertions with 4 bp TSDs within *shot* ~300 base pair apart that did not have any close matches with canonical TEs in FlyBase or RepBase [35], but were found to match a newly discovered TE in *D. melanogaster* in the *gypsy* superfamily called *Kuruka* [36]. Similar to findings in *Nipped-A*, the third allelic subline carried a large 379 base pair deletion in a separate exon. A single sequenced lethal line from 2018 (that carried a lethal premature stop codon in drosha) carries the *Kuruka* insertion in the identical location in shot. The subline with the deletion is also allelic to 4 other sublines that mapped to the 60kb region containing *shot* but did not map directly to *shot* using our knock-out line (Fig 5).

**Table 1. Abundant lethal allelic sets that successfully fine-mapped to a single gene.**

| Gene (% of sample) | Sublines (date collected) | Candidate mutation(s) | Genomic Location of Mutations |
|---|---|---|---|
| Nipped-A (0.9%) | June 8, 2018 | Transib1 | 2R: 5,186,457 |
|  | June 22, 2018 | Transib1 | 2R: 5,187,820 |
|  | June 25, 2018 | Transib1 | 2R: 5,186,458 |
|  | July 14, 2020 | Transib1 | 2R: 5,186,458 |
|  | June 25, 2021 | 414 bp deletion | 2R: 5,180,828–5,181,242 |
|  | July 29, 2021 | Transib1 | 2R: 5,186,250 |
| L(2)gl (1.3%) | June 8, 2018 | Telomeric deletion | 2L: ~13,500 |
|  | June 12, 2018 | Telomeric deletion | 2L:~21,000 |
|  | September 6, 2018 | Telomeric deletion | 2L: ~21,000 |
|  | September 24, 2018 | Telomeric deletion | 2L: ~24,000 |
|  | June 2, 2021 | Telomeric deletion | 2L: ~16,000 |
|  | June 27, 2021 | Telomeric deletion | 2L: ~21,000 |
|  | July 5, 2021 | Telomeric deletion | 2L: ~14,000 |
|  | July 10, 2021 | Telomeric deletion | 2L: ~10,000 |
| Drosha (0.9%) | July 17, 2018 | C>G premature stop | 2R: 7,933,973 |
|  | July 23, 2018 | C>G premature stop | 2R: 7,933,973 |
|  | July 27, 2018 | C>G premature stop | 2R: 7,933,973 |
|  | July 27, 2018 | C>G premature stop | 2R: 7,933,973 |
|  | August 2, 2018 | C>G premature stop | 2R: 7,933,973 |
| Ca-alpha1D (0.9%) | June 19, 2018 | Did not sequence | |
|  | August 2, 2018 | G>T splice site donor | 2L: 16,173,545 |
|  | September 14, 2020 | Did not sequence | |
|  | May 19, 2021 | C>T premature stop | 2L:16,176,233 |
|  | June 26, 2021 | Did not sequence | |

All lethal lines represented did not complement with the knock-out line of the stated gene. Candidate mutations identified and unique to that allelic set are listed. The exact breakpoints of the telomeric breakpoints cannot be called, but are estimated based visualization of the respective haplotagged bam files.

All four sublines contained TE insertions in the same *shot* exon specific to a unique transcript that our knock-out line does not disrupt. Three sublines contain TE insertion each in the same location and fourth subline contains another lethal *Transib1* insertion 700 base pairs away (Fig 5). A different lethal line from 2018 (that carried a lethal premature stop codon in drosha) carries the *Transib1* insertion in the identical location in shot. The two candidate mutations that are present in lines that did not map to that region, suggest that either (1) the observed disruption of *shot* is not lethal and we have not found the mutation in the region causing lethality, (2) disruption of shot is lethal but only in combination with specific other alleles at other loci (hence why it was fine in some crosses), or (3) that these insertions may be either in the *CyO* balancer chromosome or have inserted themselves into *shot* after deficiency crosses were performed. A different set of 4 allelic lethal sublines (collected in 2018 and 2021) mapped to a 10kb region adjacent to *shot* and all contain a *Transib1* insertion in the gene *CG33155*, for which no knock-out lines were available. The three sublines collected in the same year carried *Transib1* at the same base pair and appear to be related (Fig 4) and the other subline (collected 3 years later) carried *Transib1* inserted 39 base pairs away in the same exon (Fig 5).

Complete *Transib1* consensus sequences were determined for 7 lethal insertions (5 lethal *Nipped-A* insertions, the potentially lethal *shot* insertion, and 1 of the 3 related 2018 *CG33155* insertions—we did not include all three because we did not want the close kinship of the flies to bias the results). The 2021 insertion in *CG33155* was identified via Illumina

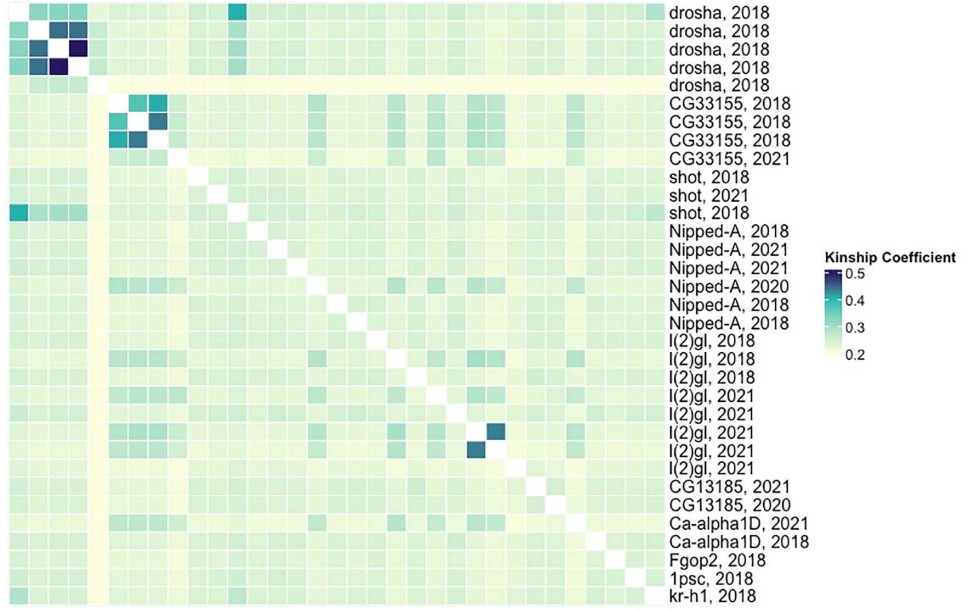

**Fig 4. Kinship coefficients between lethal lines with candidate lethal mutations.** The heatmap below shows kinship coefficients (calculated from second chromosome variants) between each lethal line that successfully fine-mapped to a single gene. The order of the rows and columns is the same from left to right and top to bottom, and lethal lines are clustered by allelic set (if present). The row is labeled with the gene with candidate lethal mutation and the year the wild fly was collected. The middle diagonal boxes are white (NA) because they represent the kinship coefficient between a lethal line and itself. Kinship coefficients do not drop much below 0.2 because all sequenced lines carry the same CyO balancer chromosome. Underlying data can be found in Table D in S1 Data.

**Table 2. Lethal mutations identified in hotspot region on 2R.**

| Gene (% of sample) | Sublines (date collected) | Candidate mutation(s) | Genomic Location of Mutations |
|---|---|---|---|
| *shot* | August 16, 2018 | 379 bp deletion | 2R: 13,879,851–13,880,230 |
| | September 24, 2018 | *Kuruka* insertion | 2R: 13,868,680 |
| | July 25, 2021 | *Kuruka* insertion | 2R: 13,868,318 |
| CG33155* (0.7%) | May 28, 2018 | *Transib1* | 2R: 13,954,314 |
| | June 22, 2018 | *Transib1* | 2R: 13,954,314 |
| | July 29, 2018 | *Transib1* | 2R: 13,954,314 |
| | June 24, 2021 | *Transib1* | 2R: 13,954,353 |

All shot lethal lines represented did not complement with the knock-out line. Four other sublines that did not map to shot are allelic with the subline collected on August 16, 2018, but they complemented with our shot knock-out line so are not represented in the table.

*CG33155 was identified through the best candidate mutations in sequence data (no knock-out line was available for this gene to confirm via crossing).

short read sequencing, thus the entire consensus sequence could not be determined. Of the 6 lethal insertions, 4 of them are 100% identical with one another and a recently described *Transib1* sequence [26]. The remaining two (both *nipped-A* insertions from flies collected in 2018) were 100% identical except for each contained a large internal deletion (~1,800 bp) (see S1 and S2 Files for sequences and alignment). To investigate sequence similarity between *Kuruka* insertions, we attempted to generate consensus sequences, but many appeared to be incomplete likely given *Kuruka*'s length (>8kb) and repetitive nature. We used soft-clipped reads from the sequence alignment for the *Kuruka* insertions that appeared in

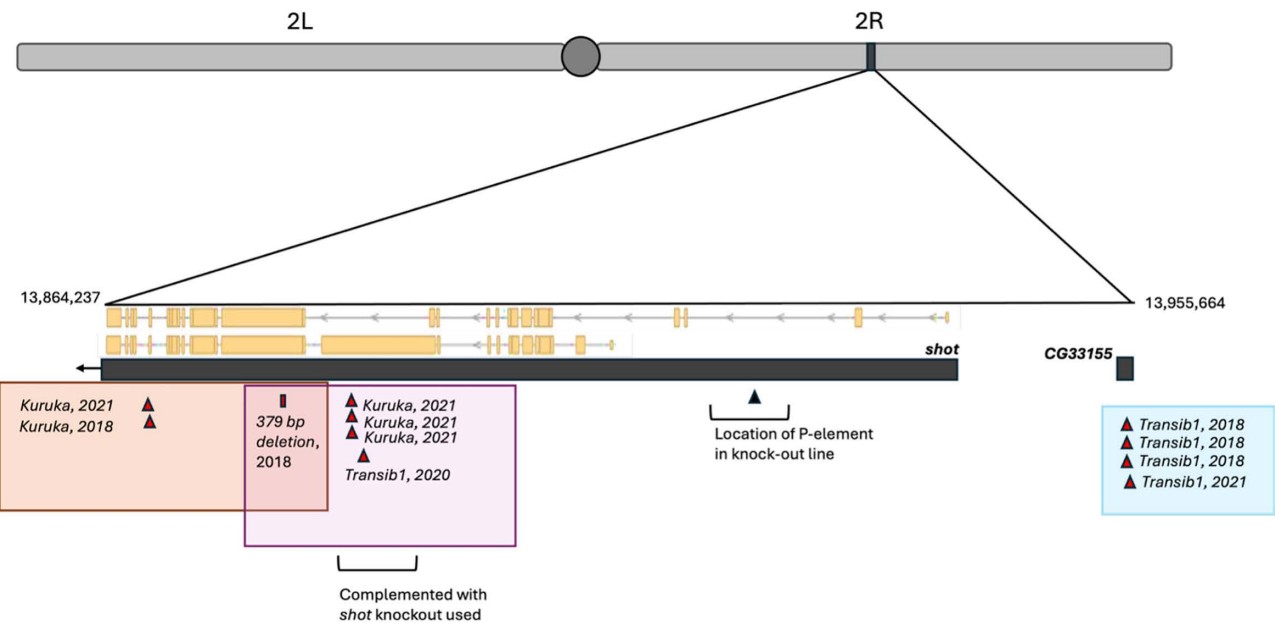

**Fig 5. Lethal hotspot region on 2R with complex overlapping allelic sets.** Mutations in 11 lethal lines that all mapped to the same deficiency are depicted in red triangles (TEs) or rectangles (deletions) above. The year the wild fly was collected is also shown. The yellow transcripts show two distinct transcripts of *shot* (yellow boxes indicate exons). Mutations present in lethal lines that compose allelic sets are surrounded by different colored boxes. There are four transposable elements in *shot* in lethal lines that are allelic to the lethal line with the deletion (pink box); however, while those four line did map to the ~60kb region that *shot* falls within, they did complement with our *shot* knock-out line. There are no other strong candidate lethal mutation in that region. These four mutations are not represented in Table 2 because we have not definitively confirmed that these mutations lethally disrupt *shot*.

the two sublines that mapped to *shot*, the sublines that mapped to *kr-h1* and that mapped to *CG18304*. Only one complete consensus sequence of *Kuruka* (that was generated and identified by PBSV) was determined from a *Kuruka* insertion in the allelic set that appeared to disrupt *shot*, but complemented with the *shot* knockout available. Alignments between all full and partial *Kuruka* sequences and a *Kuruka* sequence from Pianezza and colleagues [36] are provided in the S3 and S4 Files.

Presumably, the candidate lethal mutations we identified in balanced lethal stocks were present in the wild-caught fly and did not arise while the balanced stocks were maintained in lab. To assess this hypothesis, we selected a subset of lethal mutations and confirmed their presence in the wild fly that originated that lethal line. PCR and Sanger sequencing confirmed the presence of the *Transib1* insertion in *Nipped-a* in a wild male fly (collected July 29, 2021) and confirmed the presence of the pre-mature stop codon in *drosha* in 3 of the original wild-caught female flies in that allelic set. To test for the presence of telomeric deletions in the wild flies that originated the *l(2)gl* lethal lines, we used Illumina WGS sequencing and calculated relative read depths between the peri-telomeric region that would be heterozygous for a deletion (2L: 5,000–11,000) and a control region in the middle of the chromosome arm (2L: 10,000,000–12,000,000). If a deletion is present, we expect the read depth of the peri-telomeric region to be lower than the read depth of the middle arm and that ratio to be lower than that of flies that did not map to *l(2)gl*. We find that all 3 experimental flies we tested and which fail to complement with the *l(2)gl* knockout have lower ratios that of the two control flies (that did not map to *l(2)gl*) (Table 3).

## Genetic basis of singleton and low-frequency lethal alleles

We also map and sequence a subset of singleton (or low frequency) lethal alleles (Table 4). All disrupted genes had TE insertions as their best candidate disruptive mutation. One lethal line mapped to *Fibroblast growth factor receptor 1*

**Table 3. Wild fly read depths from peri-telomeric and control regions.**

| Wild Fly Collection Date | Did lethal line map to *l(2)gl*? | Read Depth 2L: 5,000–11,000 | Read Depth 2L: 10,000,000–12,000,000 | Read Depth Ratio |
|---|---|---|---|---|
| July 5, 2021 | Yes | 75.85 | 125.68 | 0.60 |
| July 10, 2021 | Yes | 65.95 | 137.34 | 0.48 |
| June 2, 2021 | Yes | 71.34 | 136.07 | 0.52 |
| May 20, 2021 | No | 132.31 | 127.95 | 1.03 |
| July 24, 2021 | No | 113.81 | 118.00 | 0.96 |

Read depths from Illumina WGS data from single-fly sequencing of original wild flies is presented above. The region on 2L: 5,000−11,000 represents the region that we expect the wild fly to be heterozygous for a deletion.

*oncogene partner 2*, (*Fgop2*) on 2L and the candidate mutation was found to be the same DNA transposon, *Transib1*, as found in abundant lethal genes *Nipped-A* and *CG33155*. *Transib1* appears to be responsible for multiple other lethal mutations, one in *CG13185* and another in CG18304 (found in a sublines collected in 2020 and 2018, respectively). A *Kuruka* insertion lethally disrupts CG18304 in a different exon ~1,000 bp away the lethal *Transib1* insertion in the previous line. Further, *Kuruka* insertions in the same location cause lethal affects in *Krüppel homolog 1 (Kr-h1)* in two different sublines collected in 2018 and 2021. The last singleton mapped and sequenced has a lethal TE disrupting the gene *Posterior sex combs* (*Psc*), most closely matches a Micropia-like TE insertion [37].

Complete *Transib1* consensus sequences were determined for the three lethal insertions (S1 and S2 Files). The 2021 insertion in *CG13185* is 100% identical the complete *Transib1* sequences in our data and previously described *Transib1* sequence [38]. The insertions in *Fgop(2)* and *CG18304* are 100% identical with the previous *Transib1* sequences except each contained large internal deletions (~1,500 bp and ~200 bp, respectively).

## Does the distribution of lethal mutations correspond to any features of the chromosome landscape such as gene density, recombination rate, or repetitive element density?

We investigated if there are correlations between lethal mapping distribution (controlling for deficiency size and number of deficiency crosses completed), recombination rate, and coding sequence density (Fig 6). There was no significant correlation recombination rate and mapping frequency. We also found no correlation between average GC content and lethal mapping distribution. The positive correlation between coding sequence density and lethal mapping frequency trended

**Table 4. Singleton (or low frequency) genes disrupted by lethal mutations.**

| Gene | Sublines (listed date collected) | Candidate mutation(s) | Genomic Location of Mutations |
|---|---|---|---|
| *Fgop(2)* | June 21, 2018 | *Transib1* | 2L: 6,944,237 |
| *Kr-h1* | September 22, 2018 | *Kuruka* TE insertion | 2L: 6,083,592 |
| | July 29, 2021 | *Kuruka* TE insertion | 2L: 6,083,590 |
| *1psc* | June 23, 2018 | *Micropia-like* TE insertion | 2R: 12,980,117 |
| *CG13185* | August 26, 2021 | *Doc* | 2R: 11,876,284 |
| | July 13, 2020 | *Transib1* | 2R: 11,882,111 |
| CG18304 | September 24, 2018 | *Transib1* | 2L:6,938,276 |
| | June 5, 2021 | *Kuruka* TE insertion | 2L:6,937,234 |

All lethal lines represented did not complement with the knock-out line of the stated gene with the exception of CG13185, which was identified through best candidate mutation in sequence data (no knock-out line was available for this gene to confirm via crossing).

PLOS Biology

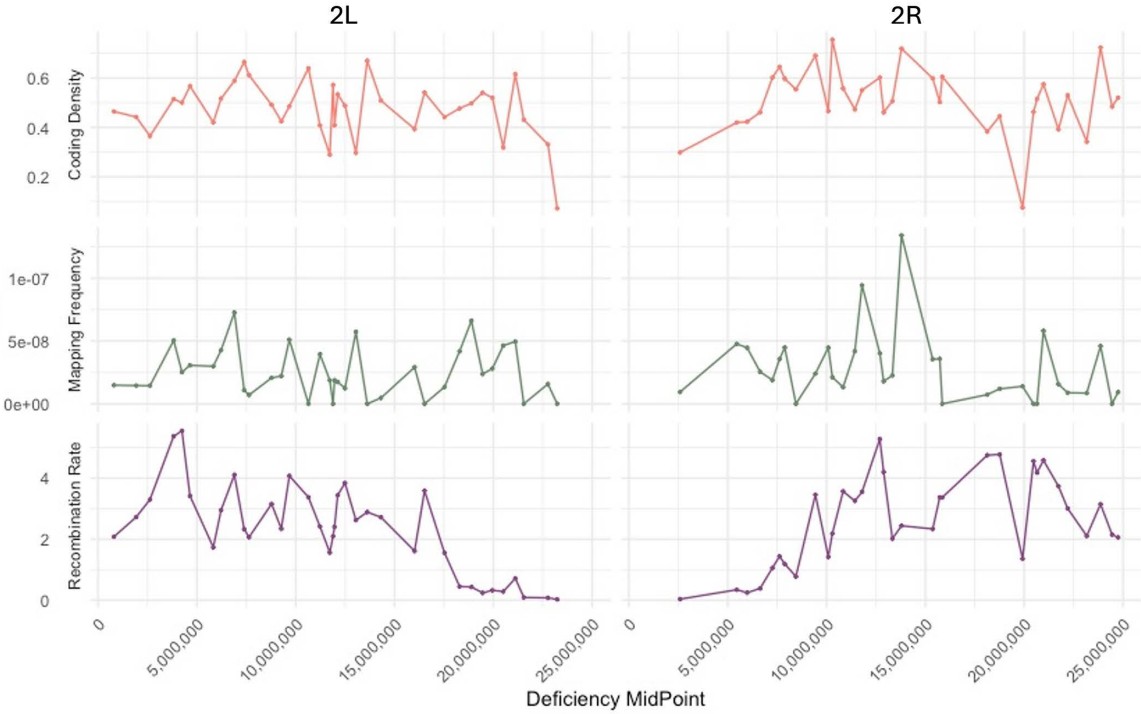

**Fig 6. Coding sequence density, lethal mapping frequency, and recombination rate (cM/Mb) distribution by deficiency.** Recombination rate estimates (cM/Mb) from Comeron, 2012 [39] were averaged per deficiency. Mapping frequency is calculated by the number of mutations mapped per deficiency divided by the deficiency size and number of crosses completed with that deficiency. Coding sequence density is calculated as the proportion of total nucleotides per deficiency that are coding. All variables are at the mid-point of each deficiency along the x-axis. The terminal deficiency that contains l(2)gl is not included in this plot because the exact size of the deficiency cannot be determined. Underlying data can be found in Table E in S1 Data.

toward significance ($r = 0.21$, $p = 0.08$). This could be explained by mutations to be more likely to be lethal when disrupting a coding region.

## Discussion

By performing chromosome-wide mapping and sequencing of hundreds of naturally occurring lethal mutations, we generate novel evidence in support of the hypothesis that most lethal chromosomes that segregate in wild *D. melanogaster* are caused by single-locus and LOF mutations and unveiled 11 different genes that carried at least one lethal disruption in our sample. Several genes were found to carry distinct lethal mutations in multiple flies (~1% or higher) sampled across all three years, suggesting that some genes are more susceptible to lethal lesions than others. Furthermore, many LOF lethal mutations are the result of TE insertions, and one TE was found to be a frequent lethal offender in both high-frequency allelic sets and singletons. The gene that exhibited the highest frequency of lethal lesions across the sample is the result of a mysterious telomeric deletion, which was seen over 40 years ago in both Russian and Californian populations of *D. melanogaster* but not tested for since. Each of these results will be discussed in turn.

By localizing hundreds of lethal mutations to small regions of the second chromosome, we provide the first direct evidence to support that most lethal chromosomes are caused by single-locus, LOF mutations. While it has been assumed that lethal alleles are primarily single locus [40], this assumption has never been directly tested and few naturally occurring lethal alleles in Drosophila have been characterized to the gene and sequence level. If many lethal chromosomes were the result of multigenic "synthetic" lethals [41,42]) or were gain-of-function, deficiency tests would be unable to locate them

and we would expect to have mapped many fewer than expectations given by the Poisson model. Given that we located as many lethal mutations as expected if they were all 'mappable' (single-locus and LOF), we provide evidence that many homozygous lethal chromosomes are due to single mutations resulting in LOF mutants. Further evidence for this claim is provided by the subset of lethal mutations that were fine-mapped to single genes and by the single, candidate mutations found in those lines.

We fine-mapped and sequenced a subset of lethal lines to discover 11 different genes with lethal disruptions, 4 of which had lethal mutations present in flies that made up ~1% or more of our sample and one, more complex, lethal "hotspot region." These high-frequency allelic sets in our sample were of primary interest for sequencing to determine if mutations that break the same genes are identical, providing a candidate site for balancing selection, or distinct, suggesting that some genes are particularly "fragile" and susceptible to lethal disruption. Our results reveal than most naturally segregating lethal mutations that disrupt the same gene are not identical-by-descent unless due to sampling bias of related individuals, and that many are the result of TE insertions, indicating that some genes are particularly susceptible to certain TEs, which is consistent with the notion that many TEs show their own specific insertion site preferences [43] . Alternatively, some sites or alleles (e.g., those that more recessive) may be under weaker purifying selection than others [44]. Our results could also explain the lack of overall trends in lethal distribution with recombination rate or GC content. While five wild flies carried identical pre-mature stop codons in *drosha*, this likely is due to stochastic demographic effects in the sample, since 4 of the 5 chromosomes bearing them appear to be more closely related than most of the chromosomes sampled.

TEs make up considerable proportions of genomes across taxa: 45%–69% of human genomes and ~20% of the *D. melanogaster* genome are composed of these mobile genetic elements [45,46]. The ambulatory nature of TEs gives them unique mutational ability. TE transposition rate varies between types and is highly variable between populations, but estimated to be on the order of $10^{-5}$ per copy per generation in *D. melanogaster* [47]. Their insertions and deletions often lead to deleterious effects for their host genome, resulting in an antagonistic co-evolution between TEs and their hosts [48]. Many animal genomes, including *D. melanogaster*, have evolved piRNA silencing pathways to reduce mobilization of TEs. Over time, immobilized deleterious TE insertions are then removed by selection and many of those that remain are likely neutral and their sequences degrade over time [49]. One way in which TEs continue to alter genomes is through horizontal transfer to a new host that is not necessarily equipped to silence them. In fact, one of the most well-studied TE transfers was the P-element, which invaded wild *D. melanogaster* from *D. willistoni* between the 1950s and 1970s [50,51].

Both *Transib1* and *Kuruka* are estimated to have recently invaded *D. melanogaster* populations 5 years or less before we began wild fly collections in 2018. *Transib1* is a cut-and-paste DNA transposon that is thought to have recently invaded *D. melanogaster* populations between 2013 and 2016 through horizontal transfer from the sister species, *Drosophila simulans* [26,52]. Originally found in 2,012 in a single sample collected in Virginia, United States, *Transib1* quickly spread worldwide from a few samples in North America and Europe in 2014 to presence in all sampled populations in 2016 (Europe (collected by Dros-EU), North America (collected by Dros-RTEC) and worldwide (collected by DGN) [53]. The exact match in the lethal *Transib1* insertion sequences in our data to one another and to that discovered by Pianezza and Scarpa, and colleagues [26] are consistent with the idea that these lethal *Transib1* insertions recently invaded *D. melanogaster*. The other TE present in the lethal 'hotspot' and in a singleton is known as *Kuruka*, a recently discovered endogenous retrovirus, and is part of the *gypsy* superfamily. This superfamily of long terminal repeat (LTR) retrotransposons, which copy-and-paste via an RNA-intermediate, but in *D. melanogaster, Gypsy* has been found to have infectious properties and be more structurally and behaviorally like vertebrate retroviruses than to other LTR retrotransposons [54]. A previous study demonstrated horizontal transfer of *Gypsy* through putting ground flies with *Gypsy* in the food of flies without *Gypsy* [55]. *Kuruka*, specifically, began invading *D. melanogaster* (from a population in Africa, most likely *D. erecta*) in 2017 and was still actively invading *D. melanogaster* populations as of 2021 [36]. We hypothesize that, because of

the recency of both TE invasions, wild populations of *D. melanogaster* had not evolved a way to immobilize *either*, which would allow both to take the new host genome by storm and repeatedly insert themselves at specific target sites with little inhibition [36].

The invasion of TEs via horizontal transfer and their impacts on lethal frequencies could extend beyond specific genes with high incidence of lethal mutations, but also the high overall frequency of lethal chromosomes. TEs (with a predominance of *Transib1* and *Kuruka*) were the most likely lethal culprits in both a high-frequency allelic set (*Nipped-a*) and the lethal-hotspot on 2R (*shot* and *CG33155*) and are also the best candidates for seven of the low-frequency or singleton lethal chromosomes that we fine-mapped and sequenced. *Transib1* appeared in three different lethal lines, causing lethal disruption in *fgop(2)*, *CG13185* and *CG18304*. The novel *Kuruka* insertion appeared in *Kr-h1* in two different lethal lines collected 3 years apart. The other two low-frequency lethal mutations were caused by the non-LTR retrotransposon, *Doc*, and the novel *Micropia-like* element recently sequenced in 2019 is thought to have recently invaded the original wild-caught Drosophila Genome Reference Population population of Raleigh, NC, USA [37,38]. Previously reported estimates of the frequency of recessive lethal second chromosomes in wild populations of *D. melanogaster* range from 9% to 50%, with a median of 25%. Our estimate, 47%, only falls lower than one estimate of 50% from a wild population in Raleigh, NC in 1970 [13,56]. Interestingly, our sample was collected recently after an invasion of *Transib1*, and this 1970 sample was collected relatively soon after *P-elements* were estimated to have invaded between 1950 and 1970 [50,51]. The high lethal frequency from 1970 was not likely inflated due to P-element hybrid dysgenesis because the wild females (who would carry proper P-element silencing piRNA) were mated with lab males. This raises an intriguing evolutionary question: does recency of newly invaded TEs correspond to higher overall frequencies of lethal-bearing chromosomes?

TEs are also involved (in a different way) with the most common lethal mutation in our sample, large telomeric deletions of 2L that extend 13–25kb into the reference sequence, partially or completely deleting the gene *l(2)gl*. Drosophila telomeres are different from most other eukaryotes in that Drosophila do not encode *telomerase*, but instead their telomeres are composed of a suite of three non-LTR retrotransposons, *HeT-A*, *TART*, and *TAHRE* that perform the function of chromosome elongation and maintenance [57,58]. So, these lethal lesions involve TE deletions rather than insertions. The estimated breakpoints of these deletions differ, but because chromosome ends (especially those that lack telomeres) degrade over time, we cannot detect if the deletion is identical by descent or if these are due to recurrent deletion events without further analysis. Over the past 50 years, intermittent and curious findings of 2L terminal deletions in wild and lab stocks have arisen. High frequencies (1%–2%) of wild chromosomes that did not complement with *l(2)gl* deficiency stocks were found in the 1970s in both the U.S.S.R. and California, USA, accounting for over 6% of all lethal second chromosomes [59,60]. Some studies suggested a heterozygote advantage to reduced *l(2)gl* production under temperature stress, proposing the possibility that these terminal deletions could be maintained by heterozygote advantage [59,61], while another study found that certain genetic elements (historically known as male recombination (MR) elements) could induce MR in Drosophila and were associated with terminal deletions of 2L and *l(2)gl* [60]. The phenomenon of MR was later tied to the broader phenomenon of hybrid dysgenesis caused by non-reciprocal crosses between wild males and lab females (that did not carry P-elements) [60,62,63]. We are unable to find any studies within the last 50 years directly studying the connection between P-element-induced hybrid dysgenesis and recurrent terminal deletions. More recent studies have found unintended terminal deletions of 2L in lab stocks, including the BDSC chromosome 2 deficiency kit lines and that some balancer chromosomes can carry terminal deletions [64,65]. Given that balancer chromosomes are more likely to recombine near the distal ends which are further from their inversion breakpoints, there is the possibility that lethal lines maintained in balanced stocks received the terminal deletion from the balancer chromosome rather than originating in the wild. Our wild-fly sequencing results suggest, however, that these high-frequency lethal, terminal deletions do exist in wild-caught flies and are not merely an artifact of the balancer chromosomes that maintain them. Given that lethal telomeric deletions have been maintained at over 1% frequency in wild *Drosophila melanogaster* populations for over 40 years, future investigation into evolutionary mechanisms and possible involvement of hybrid dysgenesis is warranted.

Our results support the hypothesis that many naturally segregating lethal mutations are due to single, loss-of-function, TE insertions, the majority of which recently invaded *D. melanogaster*. Some lethal mutations appear to be more common than may be expected via mutation-selection balance using overly simplistic expectations of mutation rate via single-nucleotide changes. Such mutation rates are inappropriate to use to estimate equilibrium frequencies of deleterious TE insertions, and in fact, mutation-selection balance, which predicts a long-term equilibrium, may be an inappropriate model to explain these high-frequency lethal insertions in nature. *Transib1* and *Kuruka* (which only appeared within the last 10 years in D. melanogaster) must be distinct from the lethal mutations studied in the mid-19th century and we predict will not stay at such high frequencies if and when D. melanogaster evolves a silencing mechanism. Average chromosome-wide mutation-rate estimates have historically focused on single-nucleotide changes and inherently assume an unrealistic constant mutation rate along the chromosome and between mutation types. Determining quantitative expectations of lethal mutation frequencies if they are due to TEs is challenging given that TE mobilization rate estimates are limited, differ between TE type and recency of invasion of the host species, and lack of standard unit of measure for TE mobilization, none of which are easily comparable to other mutation rate units. That being said, TE mobilization rates in Drosophila are typically estimated to be on the order of $10^{-3}$ to $10^{-5}$ events per copy per generation, multiple orders of magnitude higher than $10^{-9}$ single nucleotide changes per base per generation [20,22,66–71]. Given our results and the high potential mutability of TEs, it is possible that chromosome-wide mutation-rates underestimate mutation rates of certain TEs with insertion site preferences that can result in high insertion rates in specific regions (that perhaps vary temporally depending on the recency of TE invasion). This highlights the need for empirical estimates of mutation rate landscape by genomic location and mutation type.

Determining the genetic basis and evolutionary forces behind naturally occurring lethal mutations has been a topic of debate among evolutionary geneticists dating back almost a century. Conversation surrounding the impact of balancing selection versus the rate of new mutations on lethal mutation frequencies in nature was at its highest in the 1930s through 1970s, coinciding with a time in which population geneticists were focused on point mutations as the drivers of evolution [72]. While TEs were discovered during this time, it was not until studies on lethal mutations decelerated in the 1970s, that the impacts of TE mobilization in *D. melanogaster* were discovered [63,73]. Thus, coincidentally, peak discussion of lethal allele frequencies in nature and understanding of TE activity, did not intersect. Fascinatingly, the proximate basis of lethal genetic variants studied in the 1930s through 1970s differs from that of nearly all the lethal genetic variants studied today given the recency with which the disrupting elements found here invaded the species. We might speculate that a subset of those lethal mutations resulted from the then–recent invasion of p-elements [51,52].

We report candidate lethal mutations from a sample of mapped lethal effects in a population of *D. melanogaster*. While multiple examples of rare and common lethal effects appear to be due to TE insertions in our data, we recognize that this is a small sample from a single population and do not argue that TEs are necessarily the main driver of lethal mutations ubiquitously. We also emphasize that our results are correlative, and not causative, and further investigation is warranted to prove the causative lethal effects of the candidate mutations identified in our study. The TE insertions that we report as candidate lethal mutations are shared within and unique to the mapped region in each allelic set, and we did not find any other shared, unique variants within each set. Our findings highlight the need for more studies testing if TEs result in a significant proportion of naturally occurring lethal effects and the impact of recent TE invasions on the frequency of new lethal mutations.

With these caveats in mind, we propose a possible model to explain the maintenance of a meaningful proportion of lethal genetic variation in wild populations of *D. melanogaster*. After a new TE invades the species, it rapidly spreads without host suppression and is the major cause of new lethal mutations. Over time, *D. melanogaster* evolves suppression mechanisms, allowing for natural selection to more effectively remove deleterious TE insertions and decrease overall and individual lethal mutation frequencies. Upon invasion of a new TE, the process repeats. This cyclical model would result

in lethal disruptions whose genetic basis and genomic locations will vary over time depending on the recency and class of TE invasions.

Here, we emphasize that horizontal transfer for TEs and their ability to behave as retroviruses, makes them particularly threatening to the genetic load and genomic stability of *D. melanogaster* and many other organisms. We demonstrate that several lethal mutations detectable in natural populations are due to TE insertions, and that recent invaders of *D. melanogaster* populations are responsible for a substantial proportion of the lethal mutations detected. We propose the hypothesis that the recency of TE invasions may contribute to higher overall frequencies of lethal chromosomes. Hence, genome-wide lethal-causing gene variant frequency, at least in *D. melanogaster*, may be cyclical with phases of new TE invasion, increasing abundance of lethal-causing mutations, TE suppression, and selection slowly eliminating the remaining lethal insertions.

## Materials and methods

### *D. melanogaster* maintenance

All flies were kept on Archon Scientific molasses food in narrow vials and expanded for crosses in Archon Scientific molasses food in bottles, with the exception of any unhealthy lines which were reared on German Food from LabExpress, LLC, for extra nutrients. All wild lethal stocks were reared at room temperature (20 ° C) and backup stocks maintained in a cold room at 18 ° C. Deficiency stocks and all crosses were stored at 25 ° C to ensure phenotype visibility (the curly wings phenotype from *Cy* is best expressed at 25 ° C).

### Lethal line generation from a natural population

As practiced by previous Drosophila researchers, recessive lethal chromosomes are defined as a homozygous chromosome with relative fitness less than 10% of the balancer heterozygote [13]. *D. melanogaster* males were sampled from a wild population in Durham, NC, from April through September 2018, June through October 2020, and April through September 2021. Lethal lines are generated from a single wild male put through a series of crosses utilizing balanced second chromosomes, which carry dominant phenotypes, are homozygous lethal and suppress recombination, allowing the isolation and expansion of an intact wild chromosome. The original wild-caught fly was also frozen at −80 °C for sequence analysis if needed. The crossing scheme is outlined below (Fig 7). Each lethal line is then expanded and maintained in a balanced stock. If a single fly (at generation 2 in Fig 7) spawned multiple lethal sublines (beginning at generation 3 in Fig 7), we then conducted allelism crosses between sublines generated from the same fly. If the sublines are not allelic (did not carry the same lethal chromosome), we treated them as separate lethal chromosomes. If they are allelic (contain the same lethal chromosome), we only kept one subline.

The *CyO/Sna* stock was outcrossed for two generations to wild flies to incorporate P-elements and prevent hybrid dysgenesis in the offspring [25]. Some wild flies collected in 2018 were conducted using wild female flies crossed with *CyO/Sna* males to increase our sample size before the P-element *CyO/Sna* stock was ready for crossing. The crossing scheme for wild female flies is the same as in Fig 7 above except a female offspring of the wild female was mated to a male *CyO/Sna* fly in generation 1 and a single *CyO/+* female offspring was selected for generation 2. All males were removed soon after wild-fly collection from sample vials that carried male and female flies to prevent males mating with multiple females before they were sorted into individual vials.

### Estimation of observed frequency of lethal second chromosomes

To estimate the fraction of lethal second chromosomes in the population, we divide the number of distinct lethal chromosomes by the total number of distinct wild chromosomes tested. However, because we sampled typically 1–2 sublines (fewer than 5% of wild flies had more than 2 sublines tested) descending from each male at generation 2 in Fig 7, the sample size of wild chromosomes assayed is not intuitive. We count the total number of chromosomes tested according to

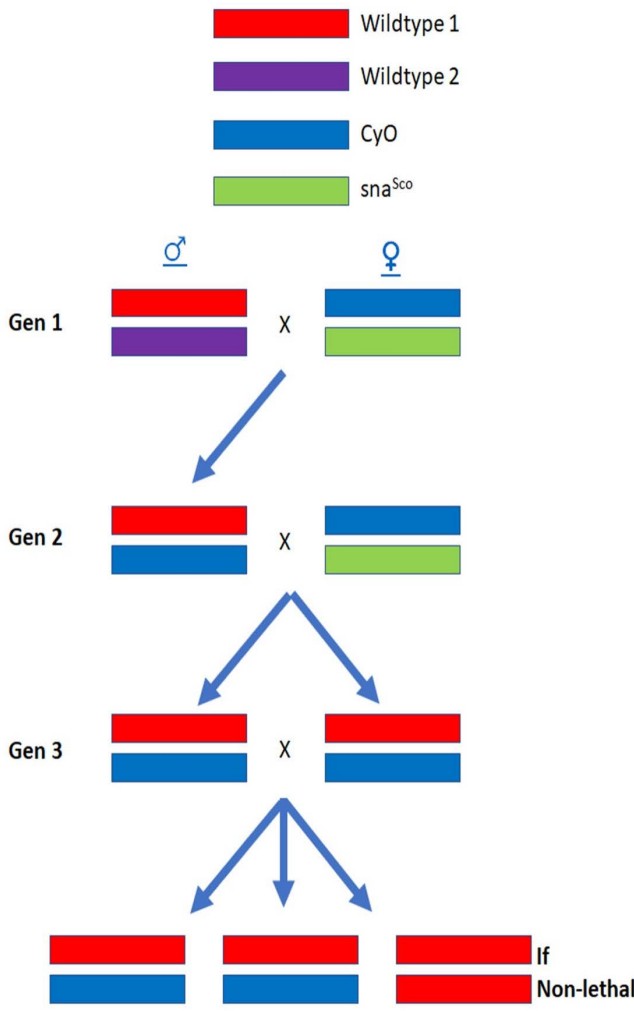

**Fig 7. Crossing scheme to determine lethality of wild second chromosomes.** Wild males, which lack crossing over, are first mated to Curly/Scutoid (CyO/Sna) virgin females (Gen 1). CyO and Sna both produce dominant visible phenotypes. The homologue carrying CyO is a balancer chromosome, which is homozygous lethal and suppresses recombination through a series of overlapping inversions. A CyO/+ male offspring is selected from the first cross, isolating a single wild second chromosome against the CyO balancer (we selected multiple male offspring from each wild fly, each creating its own subline of a potentially different sample of wild chromosome). Individual male offspring are crossed to CyO/Sna virgin females (Gen 2), and male and virgin female CyO/+ offspring are collected, as they all carry the same wild second chromosome. Generally, one or two pairs of siblings are mated (Gen 3). If the wild second chromosome is homozygous lethal, all offspring from the sibling cross will have curly wings, and a lethal subline is identified. To be labeled lethal, there must be fewer than 3.3% wildtype flies, and in practice, at least 35 curly-winged offspring and 0 straight-winged offspring.

the following rules: (a) All flies with only 1 subline (chromosome) assayed (whether lethal or nonlethal) count as 1 chromosome tested, (b) All male flies with 2 sublines completed and with 1 lethal chromosome and 1 nonlethal chromosome count as 2 distinct chromosomes tested. (c) All male flies with 2 sublines tested and neither are lethal, we assign a minimum of 1 and maximum of 2 chromosomes tested, resulting in an average of 1.5 (because we do not know if we sampled one or both chromosomes from the wild fly and that is equally likely). (d) All male flies with 2 sublines tested, and both are lethal, we conduct an allelism test. If they are allelic, only 1 distinct chromosome has been sampled (and one of the allelic lines was discarded). If they are not allelic, 2 distinct chromosomes have been sampled and both are treated as separate lines. (e) Multiple sublines from the same female wild fly are not necessarily distinct because of recombination in the wild

female fly. To circumvent bias in frequency calculations, we randomly choose 1 of the sublines from each wild female fly to include and one to discard from the numerator and denominator of all frequency calculations.

## Deficiency mapping to localize lethal effects

We used a classic deficiency complementation test design to localize potential lethal mutations. Each lethal line is crossed to a suite of 70 deficiency stocks carrying a second chromosome balancer and a known deletion (ranging in size from ~160kb to 1.6Mb). The 70 stocks were obtained from the Bloomington Drosophila Stock Center (BDSC) and span 73.0% of the second chromosome (73.4% of 2L and 72.5% of 2R). Breakpoints, genotypes, and Research Resource Identifiers (RRIDs) for each deficiency are provided in S1 Table. If the wild chromosome carries a loss-of-function (LOF) lethal mutation within the bounds of the stock deletion, all offspring carrying the deletion and lethal mutation will die before eclosing, and all offspring from the cross surviving to adulthood will have curly wings. To conclude that a lethal mutation has been mapped, we must have scored at least 35 curly-winged offspring and 0 straight-winged offspring.

## Compare the observed number of lethal mutations mapped to how many we expect to have mapped if all are single-locus and LOF

A null model is generated to determine a Poisson-based expectation of the number of mutations that should have mapped if all are single locus, loss-of-function. The model is based on the condition that only a portion of the second chromosome was tested (*coverage fraction*) and assumes:

1. All lethal chromosomes are lethal due to single-locus, LOF mutations.

2. Some lethal chromosomes contain multiple single-locus, LOF lethal mutations.

3. The number of lethal mutations per lethal chromosome follow a Poisson distribution.

The proportion of non-lethal chromosomes ($P_0$) is used to estimate the mean ($\lambda$) of the Poisson distribution of number of lethal mutations per wild chromosome.

$$P_0 = e^{-\lambda} \tag{1}$$

$$\lambda = -\ln(P_0) \tag{2}$$

We estimate an adjusted mean for the Zero-Truncated-Poisson (representing all chromosomes that contain at least 1 lethal mutation).

$$\text{adjusted mean} = \frac{\lambda}{1 - e^{-\lambda}} \tag{3}$$

The expected number of lethal mutations mapped is given by multiplying the adjusted mean by the number of lethal chromosomes tested ($X_1$), the proportion of the chromosome covered by deficiencies (coverage fraction)s.

$$\text{mapped}_{\text{exp}} = X_1 \times \text{coverage fraction} \times \frac{\lambda}{1 - e^{-\lambda}} \tag{4}$$

The standard error of the proportion of second chromosomes that are not lethal when made homozygous is calculated and used to calculate the 95% confidence interval ($CI_{P_0}$)of the proportion of chromosomes that are non-lethal using a normal approximation (Eqs 5 and 6).

$$\text{SE}_{P_0} = \sqrt{\frac{P_0(1-P_0)}{n}}$$

(5)

$$\text{CI}_{P_0} = P_0 \pm 1.96\,(\text{SE}_{P_0})$$

(6)

The confidence interval is transformed to a scale of expected counts by plugging Eq 2 into Eq 4. (Eq 7) And replacing $P_0$ with the confidence interval of $P_0$ (Eq 8).

$$\text{mapped}_{\text{exp}} = X_1 \times \text{coverage fraction} \times \frac{-\ln(P_0)}{1-P_0}$$

(7)

$$\text{CI\_mapped}_{\text{exp}} = X(1) \times \text{coverage fraction} \times \frac{-\ln(CI_{P_0})}{1-CI_{P_0}}$$

(8)

## Estimating individual lethal allele frequencies

All lethal lines that map to the same deletion are crossed in pairwise combinations to each other to determine if they contain a lethal mutation that disrupt the same gene (i.e., if they are allelic). If two lines bear allelic lethal-causing mutations, only balanced (curly-winged) offspring are produced. Any two or more lethal lines that bear allelic lethal mutations are referred to as an allelic set (multiple lethal lines that all contain a lethal disruption of the same gene). The frequency of a lethal gene disruption in the sample is calculated by the number of lethal lines in the allelic set divided by the total sample size of lethal chromosomes ($n = 555$). The numerator (number of lethal lines in allelic set) also excludes any lethal lines that were randomly selected for exclusion due to being one of multiple sampled from a wild female fly (for reasoning, refer to Materials and methods section on *Estimation of observed frequency of lethal second chromosomes).* Allelic sets of 5 or more lethal lines are referred to as abundant lethal genes. Five was chosen arbitrarily to represent high abundance, but it represents a frequency of approximately 1% from the natural population sampled (see Results).

## Fine-mapping deficiency crosses and P-element knock-out crosses

Lethal lines present in select allelic sets and some unique lethal lines had their lethal mutations fine-mapped to smaller deficiencies and mapped to single genes using P-element knock-out lines (details on fine-mapping and knock-out lines are provided in S2 and S3 Tables). The crossing and scoring methods are the same as described for broad deficiency crossing.

## Whole-Genome Sequencing (WGS), alignment and phasing

Short-read data was used for three lethal lines that died before we began sequencing with PacBio Revio. For these three lines, DNA was extracted using Qiagen midi-prep kit using the Qiagen Genomic DNA Handbook's Preparation of Tissue Samples Protocol and Isolation of Genomic DNA from Blood, Cultured Cells, Tissue, Yeast, or Bacteria using Genomic-tips Protocol. Illumina WGS ($2 \times 150$ bp reads) was performed by *Azenta (Genewiz)* and final coverage was 100×. The resulting sequence data was aligned to reference genome Release 6.57 following BWA-MEM and GATK best practices [74].

High molecular weight DNA was extracted from the remaining lethal lines using MagAttract HMW DNA Kit (QIAGEN). HiFi library prep (PB-LP-HiFi-2) and sequencing using PacBio Revio (PB-Rev-SMRT-30) were conducted through the Duke University School of Medicine Sequencing and Genomic Technologies. WGS data is aligned to

the *D. melanogaster* Reference Genome Release 6.57. Reference indexing and WGS alignment is performed using pbbioconda package *pbmm2* version 1.16.99 [75]. Aligned.bam files are filtered for chromosome 2 alignments and indexed using *samtools* version 1.14 [76]. SNPs and small variants were called and filtered using *DeepVariant* version 1.6.1 [77], structural variants called and filtered using *pbsv* version 2.10.0 [78] and tandem repeats were called and filtered using *trgt* version 1.4.1 [79]. *HiPhase* version 1.4.5 is used for phasing and haplotype calling from all three VCFs from each lethal line [80].

**Variant detection from using VCF and aligned bam files**

The three variant call files (with SNPs, structural variants, and tandem repeats) for each sample were merged and duplicates removed using *bcftools* version 1.4 [76]. Variants are annotated using SnpEff version 5.2c and the annotated variant call files are filtered to contain only high or moderate effect variants within the fine-mapped genomic breakpoints. Any variants that are present in the sequence data of lethal lines that did *not* map to the region of interest are also removed (this would indicate that the variant is not lethal or is present in the *CyO* balancer chromosome). In some cases, only one high-effect variant remained. In others, no high-impact variants remained, and the aligned bam files were manually investigated to look for mutations that were not called by any of the three programs (*DeepVariant*, *pbsv,* and *trgt*), namely TE insertions. TE insertions were identified in the bam files through visual identification of higher coverage of 5–10 base pair target site duplications. Geneious Prime (version 2025.0.3) was used to create consensus sequences of the inserts which were identified using both FlyBase Blast and RepBase RepeatMasker (https://www.girinst.org/censor/index.php, default options) to identify TE matches [35]. For all candidate lethal mutations identified, we confirmed that the mutation was not present in any of the other lethal lines that were non-allelic. To investigate the relative kinship values between lethal lines, we used *VCFtools* version 1.17 –relatedness2 option that uses the kinship coefficient described by Manichaikul and colleagues [81].

**Confirmation of candidate mutations in wild flies**

To confirm that the lethal mutations identified in the balanced lethal lines are present in the wild flies (and not a product of mutation accumulation within the balanced lethal line), we amplified and sequenced the candidate lethal mutations from a subset of the original wild flies. Primers were synthesized by Integrated DNA Technologies (IDT), and Sanger sequencing was conducted by Eton Biosciences. For a subset that were identified to carry telomeric deletions, we used WGS rather than PCR and Sanger sequencing to confirm the mutation in the wild flies. Single-fly DNA extractions for WGS were performed by a method adapted from Bernard Kim's protocol [82]. Illumina sequencing (NextSeq 1000 P2 XLEAP 300 cycles) was conducted through the Duke University School of Medicine Sequencing and Genomic Technologies and data was processed and aligned to the reference genome release 6.57 under BWA-MEM and GATK best practices.

**Distribution of lethal mutations relative to coding sequence density and recombination rate**

Average coding sequence density per deficiency (proportion of total base pairs that are coding) is calculated using coding sequencing data from FlyBase (FB2024_06, released December 19, 2024). Recombination rate estimates per 100 kilobases that had been lifted over to the current *D. melanogaster* reference genome version 6.57 were from Comeron and colleagues (2012) [83,84]. Data was processed using R version 4.4.2 and using Bioconductor (Release 3.20) package Genomic Ranges to convert and average recombination rates and coding sequence density between each of the 70 deficiency breakpoints. We controlled for size of deficiency and number of crosses to each deficiency in our lethal mapping frequencies to each deficiency region by dividing the number of lethal mutations by the size of the deficiency and the number of lethal lines crossed to the that deficiency.

## Supporting information

**S1 Table. Deficiency lines used for coarse-mapping.**
(DOCX)

**S2 Table. Fine-mapping deficiency lines.**
(DOCX)

**S3 Table. Gene knock-out lines.**
(DOCX)

**S1 Fig. Integrative Genomics Viewer (IGV) Phased Haplotypes of Telomeric Deletions.** The following images show the PacBio Revio reads aligned to the distal end of 2L. The two colors in the view of each subline depict reads phased to different haplotypes by HiPhase.
(DOCX)

**S1 File. *Transib1* sequences (fasta).**
(TXT)

**S2 File. *Transib1* sequence alignments (fasta).**
(FASTA)

**S3 File. *Kuruka* sequences (fasta).**
(TXT)

**S4 File. *Kuruka* sequence alignments (fasta).**
(FASTA)

**S1 Data. Supporting data for Figs 1, 2, 3, 4, and 6.**
(XLSX)

## Acknowledgments

We thank the School of Medicine for the use of the Sequencing and Genomics Core Facility for PacBio Revio WGS service. We thank Duke University School of Medicine for the use of the Sequencing and Genomic Technologies Core Facility, which provided NextSeq 1000 P2 XLEAP 300 WGS service for single wild flies.

There were multiple individuals who contributed in important ways to this work. We thank Isabel M Ott (Matute Lab, UNC) for WGS DNA extraction from single wild flies. We thank Eric C. Lai for insight on l(2)gl and supply of both *drosha* and *l(2)gl* knock-out lines. Thanks to Edwin Iverson for statistical analysis consulting. We thank the invaluable work that made this project possible from Christian Campbell, Molly Chakraborty, Emma Glenn, Samantha Gottlieb, Tara Maier, Hidaya Ougui, Nicolas Pardo, Denise Shkurovich, Kara Wall, and Emily Weil.

## Author contributions

**Conceptualization:** Sarah B. Marion, Mohamed A. F. Noor.

**Data curation:** Sarah B. Marion, Katrina Focht, Iman Hamid, Hannah John, Brenda Manzano-Winkler, Amber Navarra.

**Formal analysis:** Sarah B. Marion, Iman Hamid, Edwin S. Iversen.

**Funding acquisition:** Mohamed A. F. Noor.

**Investigation:** Sarah B. Marion, Katrina Focht, Iman Hamid, Hannah John, Brenda Manzano-Winkler, Amber Navarra, Saniya Pangare, Mehrnaz Zarei, Mohamed A. F. Noor.

**Methodology:** Sarah B. Marion, Edwin S. Iversen, Mohamed A. F. Noor.

**Project administration:** Sarah B. Marion.

**Supervision:** Sarah B. Marion, Katrina Focht, Saniya Pangare, Mohamed A. F. Noor.

**Validation:** Sarah B. Marion.

**Visualization:** Sarah B. Marion.

**Writing – original draft:** Sarah B. Marion.

**Writing – review & editing:** Sarah B. Marion, Edwin S. Iversen, Mohamed A. F. Noor.

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
