## [Editor Report · Decision Letter 0]

10 Oct 2025

Dear Dr Marion,

Thank you for submitting your manuscript entitled "Transposable elements drive much of naturally occurring genetic lethality in Drosophila melanogaster" for consideration as a Research Article by PLOS Biology. Please accept my sincere apologies for the delay in getting back to you as we consulted with an academic editor about your submission.

Your manuscript has now been evaluated by the PLOS Biology editorial staff, as well as by an academic editor with relevant expertise, and I am writing to let you know that we would like to send your submission out for external peer review.

Once your full submission is complete, your paper will undergo a series of checks in preparation for peer review. After your manuscript has passed the checks it will be sent out for review. To provide the metadata for your submission, please Login to Editorial Manager (https://www.editorialmanager.com/pbiology) within two working days, i.e. by Oct 12 2025 11:59PM.

Kind regards,

Richard

Richard Hodge, PhD

rhodge@plos.org

PLOS

---

## [Decision Letter · Decision Letter 1]

27 Nov 2025

Dear Dr Marion,

Thank you for your patience while your manuscript "Transposable elements drive much of naturally occurring genetic lethality in Drosophila melanogaster" was peer-reviewed at PLOS Biology. Please accept my sincere apologies for the delays that you have experienced during the peer review process. Your manuscript has now been evaluated by the PLOS Biology editors, an Academic Editor with relevant expertise, and by two independent reviewers.

In light of the reviews, which you will find at the end of this email, we would like to invite you to revise the work to thoroughly address the reviewers' reports.

As you will see below, the reviewers agree that the manuscript is interesting and well done, but they raise overlapping concerns with the interpretation of the results and that alternative hypotheses are not fully addressed or discussed. In addition, they raise some concerns with several overstatements and request that additional validation experiments are provided to fully support the conclusions. This includes additional evidence for the heterogeneity in TE insertion positions and quantitative analyses to support the claim that TE mutation rates explain high lethal frequencies.

Given the extent of revision needed, we cannot make a decision about publication until we have seen the revised manuscript and your response to the reviewers' comments. Your revised manuscript is likely to be sent for further evaluation by all or a subset of the reviewers.

**IMPORTANT - SUBMITTING YOUR REVISION**

*Re-submission Checklist*

*Published Peer Review*

*PLOS Data Policy*

*Blot and Gel Data Policy*

Best wishes,

Richard

Richard Hodge, PhD

rhodge@plos.org

REVIEWS:

Reviewer #1: The authors mapped recessive deleterious mutations from a natural D. melanogaster population and found that many mutations were caused by insertions of two TEs, which recently invaded D. melanogaster.

The study is thoroughly executed and the results are interesting. Nevertheless, I do not agree with the interpretation of the authors:

1) It was recently shown that variable strength of purifying selection could generate the same pattern as insertion bias (Langmuller et al. NAR 2022)- insertions are preferentially detected at specific genomic positions. Hence, it is conceivable that the genes with recessive lethal TE insertions are not very strongly counter selected because they were highly recessive. This interpretation would not only be supported by the presence of multiple independent insertions at the same gene, but also by the observation of the authors that the same gene was hit by another mutation causing a recessive deleterious allele (Nipped-A and CG33155). Hence, the results would be mostly affected by the dominance of a gene and the most recessive genes would be enriched for recessive lethals. While the reviewer considers this the most plausible interpretation of the data, the reviewer accepts the speculative nature, but requests that the hypothesis is adequately discussed in a revised manuscript.

2) If the large number of independent mutations at some of the genes does not reflect an insertion bias, but differential selection, this implies that the TE invasion has resulted in many mutations of the same gene. In this case, the reviewer is not clear about the expectations to detect point mutations or TE insertions to cause lethal mutations. Maybe, this could be modelled by comparing a mutational burst to continuously emerging base substitutions. After all, the number of genes affected by base substitutions is not so small.

Apart from the interpretation of the data, the reviewer would like to see additional evidence for the heterogeneity in TE insertion positions. Mapping reads with TE insertions to a reference genome without them is clearly challenging. Hence, the reviewer would like to see an alignment of reads spanning the insertions.

Minor points:

Isoline seems to be internal lab slang-the reviewer prefers the term isofemale line. Wikipedia thinks that an isoline is something entirely different…

Reviewer #2: The study investigates the long-standing question of why recessive lethal alleles persist at unexpectedly high frequencies in natural population of Drosophila melanogaster. By combining classical balancer-based complementation mapping with whole genome sequencing of 293 wild second chromosome lines, the authors demonstrate that most lethals seem single-locus, loss-of-function mutations and frequently arise independently in the same 'essential genes.' Interestingly, they find that approximately 80-90% of these mutations occur in relation to transposable element (TE) insertions. The authors propose that recurrent TE insertions, with their much higher mutation rates than single-nucleotide changes, can account for the observed frequencies of lethal alleles through mutation-selection balance. They further suggest that recent TE invasions create transient bursts of lethality that are later purged as host suppression evolves.

Overall, the manuscript reframes the evolutionary origin of natural lethality as a dynamic consequence of TE activity. However, there are some concerns that should be addressed to be sure that the conclusions are supported by the presented data.

Major Comments:

1) In general, the authors exclude alternative hypotheses to explain their data with insufficient support.

a. In the introduction the authors heavily focus on mutation-selection balance and continue to build their arguments on its inadequacies to explain the observed higher frequencies of lethal mutations present. Even though they briefly mention balancing selection (both in the introduction and discussion) they dismiss it and it is not given enough attention to rigorously test. This is also reflected in that they do not test for or present explanations for any heterozygote fitness that might be in play.

b. In addition, explanations other than TE insertions (e.g. other structural variation, demography, genetic drift) also need to be evaluated as the manuscript largely leads the reader to believe TEs are not only the major source but almost the sole driver of lethal mutations.

c. The authors should be careful when interpreting results: Failure to reject the null hypothesis is not proof of the null hypothesis being true (type I vs type II errors are different) (see lines 256-266). The conclusions should be toned down, or the null hypothesis reframed so the authors can actually perform a test to see if their hypothesis (the non null) is more likely.

2) Several strong claims are made without direct evidentiary support and should be toned down or removed if the authors do not perform additional validation analyses/experiments.

a. The claim that TE mutation rates (not only invasion) explain high lethal frequencies is not backed by quantitative parameters.

b. The finding that the majority of mapped lethal alleles coincides with TE insertions is not supported by functional data to confirm these events are indeed causal, with mere correlative evidence to support it. Thus, the statement "TEs drive much of naturally occurring lethality" goes beyond what the data shows.

3) The manuscripts central claim that high lethal mutation frequencies can be explained by mutation-selection balance (a long-term equilibrium) once TE mutations are considered presents an inconsistency with the interpretation of the findings that these TE invasions are transient. If said invasions are dynamic and result in short-term elevations which would be selected out once the species develops a silencing mechanism for the invader then the hypothesis that it is a long-term equilibrium wouldn't stand. This should be evaluated and discussed.

4) The authors report several allelic sets that show high kinship (likely due to sampling bias) but do not mention any correction for it.

5) The authors introduce 'Kuruka' as a novel Gypsy-like retrotransposon implicated in several mapped lethal alleles. Although this could be an exciting discovery, more information is required in this case, including a consensus sequence and genomic coordinates for insertions, along with phylogenic comparison with other established Gypsy-like elements, as well as its distribution in other (lab or not) D.melanogaster populations.

---

## [Decision Letter · Decision Letter 2]

27 Jan 2026

Dear Dr Marion,

Thank you for your patience while we considered your revised manuscript "Transposable elements shape naturally occurring genetic lethality in Drosophila melanogaster" for publication as a Research Article at PLOS Biology. This revised version of your manuscript has been evaluated by the PLOS Biology editors, the Academic Editor and the original reviewers.

Based on the reviews, I am pleased to say that we are likely to accept this manuscript for publication, provided you satisfactorily address the remaining points raised by Reviewer #1. In addition, please also make sure to address the following data and other policy-related requests that I have provided below (A-F):

(A) We routinely suggest changes to titles to ensure maximum accessibility for a broad, non-specialist readership. In this case, we would suggest a minor edit to the title, as follows. Please ensure you change both the manuscript file and the online submission system, as they need to match for final acceptance:

“Transposable elements contribute substantially to naturally occurring genetic lethality in Drosophila melanogaster"

(B) You may be aware of the PLOS Data Policy, which requires that all data be made available without restriction: http://journals.plos.org/plosbiology/s/data-availability. For more information, please also see this editorial: http://dx.doi.org/10.1371/journal.pbio.1001797

-Supplementary files (e.g., excel). Please ensure that all data files are uploaded as 'Supporting Information' and are invariably referred to (in the manuscript, figure legends, and the Description field when uploading your files) using the following format verbatim: S1 Data, S2 Data, etc. Multiple panels of a single or even several figures can be included as multiple sheets in one excel file that is saved using exactly the following convention: S1_Data.xlsx (using an underscore).

-Deposition in a publicly available repository. Please also provide the accession code or a reviewer link so that we may view your data before publication.

Figure 1, 2, 3, 4, 6

(C) Please deposit the whole genome sequencing datasets in a public repository and provide the accession number in the Data Availability Statement in the online submission form.

(D) Please also ensure that each of the relevant figure legends in your manuscript include information on *WHERE THE UNDERLYING DATA CAN BE FOUND*, and ensure your supplemental data file/s has a legend.

(E) Please ensure that you are using best practice for statistical reporting and data presentation. These are our guidelines https://journals.plos.org/plosbiology/s/best-practices-in-research-reporting#loc-statistical-reporting and a useful resource on data presentation https://journals.plos.org/plosbiology/article?id=10.1371/journal.pbio.1002128

- If you are reporting experiments where n ≤ 5, please plot each individual data point.

(F) Per journal policy, if you have generated any custom code during the course of this investigation, please make it available without restrictions. Please ensure that the code is sufficiently well documented and reusable, and that your Data Statement in the Editorial Manager submission system accurately describes where your code can be found. More information on our Code Policy, what and how to share can be found here: https://journals.plos.org/plosbiology/s/code-availability

We expect to receive your revised manuscript within two weeks.

*Published Peer Review History*

*Press*

Best regards,

Richard

Richard Hodge, PhD

rhodge@plos.org

Reviewer remarks:

Reviewer #1: The authors have responded to my comments, but prefer to stick to their original interpretation. I think this is perfectly ok, as they at least discuss alternative interpretations.

My only concern that remains is their weak justification that SNPs are not causative. It is well-understood that even silent mutations can have large fitness consequences-there is a large body of literature on this in bacteria. Along the same lines-intronic sequences are often very important for gene regulation. Hence, I think that some more careful phrasing is required for this aspect of the manuscript as well.

Reviewer #2: The authors have carefully responded to my previous concerns. I am satisfied that the presented conclusions are supported by the data and analyses.

---

## [Editor Report · Decision Letter 3]

4 Feb 2026

Dear Dr Marion,

On behalf of my colleagues and the Academic Editor, Mark Siegal, I am pleased to say that we can accept your manuscript for publication, provided you address any remaining formatting and reporting issues. These will be detailed in an email you should receive within 2-3 business days from our colleagues in the journal operations team; no action is required from you until then. Please note that we will not be able to formally accept your manuscript and schedule it for publication until you have completed any requested changes.

PRESS

Best wishes,

Richard

Richard Hodge, PhD

rhodge@plos.org

PLOS
